# Effects of an Acute Dose of Zinc Monomethionine Asparate and Magnesium Asparate (ZMA) on Subsequent Sleep and Next-Day Morning Performance (Countermovement Jumps, Repeated Sprints and Stroop Test)

**DOI:** 10.3390/nu16152466

**Published:** 2024-07-29

**Authors:** Ben J. Edwards, Ryan L. Adam, Dan Drummond, Chloe Gallagher, Samuel A. Pullinger, Andrew T. Hulton, Lucinda D. Richardson, Timothy F. Donovan

**Affiliations:** 1Research Institute for Sport and Exercise Sciences, Liverpool John Moores University, Liverpool L3 3AF, UK; ryan41548@gmail.com (R.L.A.); c.gallagher@2022.ljmu.ac.uk (C.G.); l.d.richardson@ljmu.ac.uk (L.D.R.); t.f.donovan@ljmu.ac.uk (T.F.D.); 2Sport Science Department, Inspire Institute of Sport, Vidyanagar, Bellary 583275, India; samuel.pullinger@inspireinstituteofsport.com; 3Department of Nutrition, Food, and Exercise Science, University of Surrey, Surrey GU2 7XH, UK; a.hulton@surrey.ac.uk

**Keywords:** actimetry, magnesium, zinc, vitamin B6, pyridoxine, CMJ

## Abstract

The goal of the present study was to determine whether an acute dose of a zinc-containing nutritional supplement (ZMA) has any effects on sleep and morning performance in recreationally trained males. Nineteen males participated in a repeated-measures within-subjects study to assess objective and subjective measures of sleep, completed counter-movement jumps (CMJ) and repeated sprint morning performance (RSP). Three days of baseline food intake showed no major deficiencies of zinc, magnesium or vitamin B6 for all participants (11.9 ± 3.4, 395 ± 103 and 2.7 ± 0.9 mg.day^−1^, respectively). Sleep (22:30–06:30 h) was assessed via actimetry, and either a control (no tablets, NoPill), dextrose placebo (PLAC) or ZMA was ingested 30–60 min before retiring to bed for two nights. The participants undertook the three conditions (NoPill, PLAC or ZMA) administered in a counterbalanced order. The data were analyzed using general linear models with repeated measures. In healthy active males who consume diets of adequate micronutrients, sleep normally and maintain good sleep hygiene (time to bed and wake times), ZMA supplementation had no beneficial effect on RSP or performance in the Stroop test (*p* > 0.05) but did improve CMJ height (*p* < 0.001) compared to that of PLAC but not NoPill (*p* > 0.05). Supplementation of ZMA for two nights had no effect on sleep, RSP or cognitive function. The NoPill condition elucidated the effects of the intervention under investigation.

## 1. Introduction

The use of ergogenic aids among athletes has become a common practice as they consistently seek to gain a competitive advantage, with reports citing use among elite athletes of up to 81–100% [1]. Of increasing interest to athletes, both recreational and elite, is the beneficial role that adequate mineral element nutrition can have on physical performance [2]. Nutritional supplement companies claim that exogenous supplementation of zinc monomethionine asparate and magnesium asparate (ZMA) promotes deep and restful sleep (due to magnesium’s ability to normalize and extend stage 3 slow-wave sleep), aid in circadian regulation and increase testosterone levels [3], thereby aiding recovery and increasing muscular strength. It has recently been proposed that the mechanism by which Zn and Mg can promote sleep is through their roles in synthesis and function of sleep–wake neurotransmitters such as gamma-aminobutyric acid (GABA)—a receptor that supports sleep when activated [4]. Sleep is widely regarded as one of the most influential modes of recovery for athletes [5,6], with a positive relationship between sleep and optimal training load proposed within the literature [7]. Even with the opportunity for 8 hours of sleep, athletes will only take ~6 hours, leading to small amounts of sleep loss [8], regardless of other factors that can reduce sleep quality and quantity, such as training times and competition [9].

Sleep loss can have detrimental effects on mood and motivation, essential elements for tasks requiring higher cognitive function (such as executive functions), especially in the morning, when it is compromised [9]. Executive functions notably include the ability to plan and coordinate considered action regardless of alternatives, as well as to monitor, update and suppress distractions by focusing attention on relevant information (i.e., inhibition). The Stroop Color–Word test is used to measure attention and notably its inhibitory processes by the ability to fully suppress the influence of a dominant source of information, such as automatic word reading [10]. The idea that ZMA improves sleep quality is extremely appealing to athletes and coaches alike, hence there is anecdotal evidence from coaches of national, international, world and Olympic athletes who consistently report that individuals, especially those participating in team sports such as rugby, are regularly consuming the supplement [11].

Zinc is an essential trace element that plays an important role in living cells of the human body, with its deficiencies being positively associated with impaired immune function and decreased performance [12]. Magnesium is a ubiquitous element that aids several physiological processes, including glycolysis and fat and protein metabolism, alongside enhancing immune, neuromuscular, cardiovascular and hormonal function [13]. Although there is little research on the impact of vitamin B6 (pyridoxine) on performance, there is some evidence that its deficit may impair protein and glucose metabolism [14] and impact cognitive function [15], while high levels of it may impair sleep [16]. Investigations into the effectiveness of Zn and Mg supplementation have focused on populations whose daily habitual intake is below the recommended levels, who are either very young or aged and have clinical conditions, such as those with insomnia [17,18]. Some methodological issues that have limited the interpretation of this area are a lack of (i) prior measurements of diets for macro- and micronutrient quantities, (ii) a baseline assessment of habitual sleep and resultant sleep after ingestion of the supplement and (iii) the absence of a “Pill”, placebo (PLAC) and experimental (ZMA) conditions. Recent work that addressed the concerns about research design has investigated whether recreational athletes (n = 16) suffering from partial sleep deprivation (who are otherwise healthy with no sleep disorders and have a balanced diet) may experience improvements in sleep variables with acute ZMA supplementation [19]. No benefits of supplementation to sleep (objective or subjectively measured) nor morning Stroop test performance have been found. Furthermore, research investigating acute ZMA supplementation (1–2 nights) is therefore warranted in populations of “normal sleepers” with balanced diets who may have sleep loss through an 8-hour opportunity of sleep to further understand its effects on sleep, morning cognitive performance and physical performance measures. One such measure is repeated sprint ability performance (RSP)—the ability to repeatedly perform maximal sprints (≤6 s) with limited recovery between bouts (≤60 s)—an important performance test that carries relevance for team-based sports but is also sensitive to environmental changes such as time-of-day [20]. 

Therefore, the aims of this study were to investigate the effects of an acute dose of ZMA (compared to those of a placebo (PLAC) or no-pill condition (NoPill)) taken before retiring for two nights (with the opportunity to sleep for 8 hours) on subsequent sleep and next-day morning physical (RSP and countermeasure jumps) and Stroop test (attention and notably its inhibitory processes) performance in a sample of familiarized and highly motivated participants with no known sleep disorders. We hypothesized that ZMA would have no beneficial effects on sleep variables or next-day morning physical or Stroop test performance in our chosen population of healthy male recreational athletes following two nights of supplementation.

## 2. Materials and Methods

Nineteen males, as identified by sex and gender, participated in the investigation. Sample size was determined using power calculation software (G*Power, version 3.1.9.6). To test the effects of ZMA vs. PLAC on sleep variables and cognitive performance using a paired *t*-test, based upon a small–moderate effect size of 0.6 for sleep quality and Stroop number with a power of 0.80 and α = 0.05, a sample of 19 participants was required. In line with the inclusion criteria, the participants were recreationally active (as classified by the Participant Classification Framework [21]), injury-free with no diagnosed sleep disorders, had not completed shift work or travelled outside the local time zone in the past month and did not take any dietary supplements at the time of the study. Self-reported food intake showed no signs of vitamin or mineral deficiencies. Baseline data of the participants are presented in Table 1. Prior to participating in the investigation, the participants were presented with an information sheet followed by a Physical Activity Readiness Questionnaire (PARQ [22]) and a written consent form. Verbal explanation of the experimental procedure was provided; this included the aims of the study, the possible risks associated with participation and the experimental procedures. The participants were assessed for circadian chronotype using the Composite Morningness/Eveningness Questionnaire [23]. The mean chronotype score on a 13–52-point scale was 35.6 ± 4.6, hence all participants were intermediate type. All participants gave their informed consent for inclusion before they participated in the study. Experimental procedures were approved by the Liverpool John Moores University Human Ethics Committee (P14SEC018), conducted in accordance with the ethical standards of the journal and complied with the Declaration of Helsinki.

### 2.1. Research Design

All participants were required to visit the laboratory on five occasions (dry temperature of 19 °C, 35–45% humidity and a barometric pressure of 750–760 mmHg, respectively). The participants completed two familiarization sessions one week prior to the experimental protocol. During the familiarizations, the participants completed (1) 3 days of baseline sleep measurements (two weekdays and one weekend) wearing an Actiwatch (MotionWatch 8, software 1.1.3 Cam‘n’Tech, Cambridge UK), (2) a 3-day food diary that was analyzed with Nutritics^®^ (V6, Co., Dublin, Ireland) analysis software (this process was conducted by a SENr registered Sports and Exercise Nutritionist), (3) 5 counter-movement jumps (CMJ; Force Platform, Kistler, UK) and (4) a repeated sprint ability (RSP) protocol on an indoor 100 m runway (10 × 20 m). All participants were familiarized with all questionnaires and the Stroop test for cognitive function [10]. Both familiarizations involved the collection of the participants’ height, mass, completion of questionnaires (Profile of Mood States, POMS [25], the Stanford Sleepiness Scale [26] and sleep questions from the Liverpool Jet Lag Questionnaire [27]) and completion of the Stroop test (see Figure 1 and the “Section 2.3” section for detail). The remaining sessions consisted of (iv) three experimental conditions involving two consecutive nights of prescribed sleeping (retiring at 22:30 and rising at 06:30) at the participants’ homes before entering the laboratory at 07:00 on the third day. Prior to bed, the participants either consumed three ZMA or PLAC capsules or took no pills (NoPill) dependent on the condition. To control for sleep (retiring time) and check whether the pills were consumed appropriately, the participants were reminded via a phone call at 22:15, and compliance was established verbally. The Profile of Mood States (POMS) questionnaire [25] was completed by each individual 30 min prior to sleep for all conditions. In total, the three ZMA capsules contained 30 mg of zinc, 450 mg of magnesium and 10.5 mg of vitamin B6 (PhD Nutrition LTD, Yorkshire, UK), and the placebo capsules were made in the department and contained maltodextrin (~1.55 g, Sport supplements Ltd. t/a BulkTM Colchester, UK). The researchers and participants were blinded to the supplement schedule, and the pills were provided in a plastic bottle with instructions to consume with water 30 min before retiring. Both ZMA and placebo were lightly dusted with maltodextrin to create a similar taste, had similar weight (0.8 g/capsule) and were of 00 size. At the end of the experiment, the order of treatment was revealed to the researchers by an author (BE). Before experimental sessions, the participants were asked to refrain from vigorous physical activity 24 h prior, during which they also had to avoid any alcoholic or caffeine-containing drinks. No food was to be consumed 1–2 h before the experimental protocol, for the morning testing session and before sleep. In the two hours before retiring to sleep, the participants were asked to refrain from watching television or using their mobile devices and were also required to consume supplements provided if they were in the ZMA or PLAC condition. To ensure recovery and washing out of ZMA between trials, there was at least a week between the testing conditions for all participants. The experimental sessions were then counterbalanced in order of administration to minimize any potential learning effects [28], with a minimum of 72 h to ensure recovery between trials. All experiments were completed between the months of October and May (autumn to spring in the UK), with the sunrise and sunset range from the start to the end of the experiment being from 05:32 to 07:27 and from 18:02 to 20:43, respectively. The individuals’ exposure to sunlight in the mornings prior to entering the laboratories was <80 lux.

### 2.2. Protocol

A schematic of the experimental protocol is shown in Figure 1. The participants reported to the laboratory fasted on the morning of each condition (07:00 h) and having abstained from physical activity and caffeine for 24 h. One of the following conditions was tested on each morning and separated by 48 h: control no-pill (CON), placebo (PLA) and ZMA. The sessions were counterbalanced in order of administration to minimize any learning effects [28]. On arrival, everyone was weighed semi-nude (Seca 704, Hannover, Germany) and completed all subjective measures of sleep [26,27], and cognitive function was assessed by the Stroop test [10]. Following this, resting blood lactate (mmol^−1^) was obtained via a finger-prick capillary sample and analyzed immediately (Lactate Pro, Arkray, Shiga, Japan).

After all pre-testing data were collected, a heart rate (HR) monitor and watch (T31 Transmitter, Polar, Finland) were fitted to each participant. The volunteers then performed a 10 min standardized warm-up and stretched individually for 5 min. After this preparation stage, lower body muscular power was measured via CMJ. Each participant performed a jump at 50 and 80% of perceived maximum effort before having three opportunities to reach their highest jump, each separated by 30 s of recovery. Flight duration of each jump was calculated and converted to centimeters through a commonly used and validated formula [29]. The participants then completed 10 × 20 m sprints on a biosynthetic runway using infrared timing gates (Brower TCi Timing system, Draper, UT, USA) with 30 s of active recovery after each sprint. Ratings of perceived exertion (RPE; [30]), thermal comfort (TC; [31]), HR (beats.min^−1^) and sprint times (s) were measured after each sprint. Blood lactate was obtained immediately post exercise.

### 2.3. Measurements

Prior to the main experimental laboratory sessions, two familiarization sessions reduced any learning effects of the Stroop test [10] (or the word/color interference test), where the participants were asked to read out their responses to words or colors for 45 s as quickly as possible. The test was filmed, and the number of errors and total amount completed were recorded and analyzed. The first sheet had text (red, blue, yellow, black and green) in black ink (naming the word test), the second sheet blocks of color corresponding to the text on the first sheet (naming the color test). From the third sheet, the participants had to read out the word (which was colored differently than the color designated by the word, e.g., the word was “yellow” but colored red; referred to as naming the color of the word test); only data from the naming the color of the word test for number completed and number of errors were used for analysis. Following two consecutive nights of sleep and normal activity, the participants rose at 6:30, arrived at the laboratory at 07:00 h, and after sitting for 30 min, completed the Profile of Mood State questionnaire (POMS [25]) and sleep questionnaires and rated sleepiness [26,27]. The POMS questionnaire measures mood states, with the participants answering 32 questions with 8 moods for how they “feel right now”. For the sleep questions, the participants ticked on a 10 cm scale their answers “compared to normal” with –5 “being less or earlier”, 0 being “normal” and +5 being “more or later”. Participants then undertook the Stroop test (Figure 1). To monitor sleep across the two consecutive nights, the participants put on an Actiwatch on their nondominant wrist in the evening prior to the ingestion of the pill (Motionwatch 8, CamnTech). The data were downloaded for analyses on the third morning when the participants arrived at the laboratory for testing. The second night of sleep was analyzed with the following variables: sleep latency (time taken to fall asleep after turning out the lights), sleep efficiency (the percentage of time asleep while in bed), actual sleep time and fragmentation index (the number of interruptions to sleep as a proportion of the total number of immobile 30 s epochs). In between experimental conditions, the participants were under normal living conditions.

### 2.4. Statistical Analysis

Statistical Package for the Social Sciences (SPSS IBM) version 29 for Windows was used. Differences between conditions were evaluated using a general linear model with repeated measures within the subject factor for “Pill” (NoPill, PLAC, ZMA; 3 levels), pre–post or distance (2 or 10 levels, respectively) and interactions. To correct violations of sphericity, the degrees of freedom were corrected in a normal way using Huynh–Feldt (ε > 0.75) or Greenhouse–Geisser (ε < 0.75) values for ε, as appropriate. Graphical comparisons between the means and Bonferroni pairwise comparisons were made where main effects were present. Effect sizes (d) were calculated from the ratio of the mean difference to the pooled standard deviation (Cohen’s *d*). The magnitude of the ES was classified as trivial (≤0.2), small (>0.2–0.6), moderate (>0.6–1.2), large (>1.2–2.0), and very large (>2.0) based on guidelines from Batterham and Hopkins [32]. The results are presented as the mean ± standard deviation (SD) throughout the text unless stated otherwise. Ninety-five percent confidence intervals (CI) are presented where appropriate as well as the mean difference between pairwise comparisons. All data are presented as means ± standard deviations or 95% CI in the text, tables and figures. Significant differences were reflected with *p* ≤ 0.05, and a trend with 0.10 > *p* > 0.05. 

## 3. Results

### 3.1. Actimetry and Subjective Sleep Variables

Table 2 shows the cohort means (± SD) for all actimetry variables in each condition, together with the statistical analysis. No significant main effect was evident for sleep latency, sleep efficiency, actual sleep time or fragmentation index (*p* > 0.05) in all conditions. Resting subjective measures of all qualitative sleep variables (via the Stanford Sleep Questionnaire [23]) were not significantly different between conditions (*p* > 0.05). In summary, regarding the aims, ZMA had no beneficial effect on objective or subjective sleep measures.

### 3.2. Performance

#### 3.2.1. Counter Movement Jump

There was a significant main effect of condition on obtained CMJ height (*p* < 0.001 see Table 2), where pairwise comparisons showed that the jump height of PLAC was significantly lower than that of both NoPill (mean difference of 1.78 cm, CI = 0.65 to 2.91 cm, *p* = 0.02) and ZMA (1.61 cm, CI = 0.40 to 2.81 cm, *p* = 0.007).

#### 3.2.2. Repeated Sprint Performance: Split Times, HR, RPE and TC Levels

There was no significant main effect of condition on sprint times (*p* = 0.052), HR, TC and RPE (*p* > 0.05). There was no main effect for sprint for split times (*p* = 0.217), although, as expected, HR, RPE and TC values increased from the first to the last sprint (Table 2). No significant differences and no interactions were observed for any variable (*p* > 0.05, see Figure 2). In summary, regarding the aims, ZMA had no beneficial effect on CMJ or sprint times compared to NoPill, although statistically significant differences were found between ZMA supplementation and PLAC.

### 3.3. Blood Lactate

Blood lactate values were not significantly different between all conditions (*p* > 0.05, see Table 2). There were significant time effects for blood lactate values (*p* < 0.005), with the lowest values at rest and the higher values post RSP (1.0 ± 0.3 vs. 5.9 ± 2.1 mmol^−1^). Blood lactate values displayed no interaction between condition and time such that profiles increased in parallel with the sprint number for all conditions (*p* > 0.05). 

### 3.4. Resting POMS Questionnaire Variables

Subjective ratings of fatigue were higher in the NoPill condition (NoPill: 6.6, PLAC: 5.6 and ZMA: 5.5 AU; Table 3). Values for happiness, calmness, fatigue and confusion were all significantly higher in the evening before each condition (*p* < 0.05). An interaction for CON*Time-of-day was found for resting subjective measures of tension and fatigue, where the rate of falling from night to morning was the highest in PLAC and ZMA, respectively, in comparison to the other conditions (*p* > 0.05). 

### 3.5. Stroop (Color–Word, Word–Color Interference Test)

Color number or error scores were not significantly different between all conditions (*p* > 0.05, Table 2 and Figure 2).

## 4. Discussion

The present study was designed to investigate the effects of an acute dose of ZMA on subsequent sleep and sprint/CMJ and Stroop test (an indicator of the efficiency of the inhibitory function) performance in the following morning in a large sample of familiarized and highly motivated male participants with no known sleep conditions. The first main finding of the study was that in the cohort (n = 19) there was no effect on objective (actimetry) and subjective measures of sleep or cognitive function across all conditions irrespective of consuming ZMA before bed. To the best of our knowledge, there is only one other study to report the effects of acute ingestion of ZMA on measures of sleep in a young (n = 16; age 20–24 years old), healthy and athletic population, which found no effect of two nights of supplementation on similar measures of sleep [19]. Consistently with the current study, they had NoPill, PLAC and ZMA conditions, although they used a partial sleep deprivation condition (4 h), unlike the current investigation, where the participants had the opportunity to sleep for 8 h.

Those studies that have investigated the effects of ZMA on sleep differ with regard to several factors that might affect observations relevant to sleep, such as population (age, fitness level, health issues related to sleep), conditions employed (such as placebo, control, etc.), blinding and counterbalancing the order of administration and conditions, in addition to the amount, timing and other active ingredients in the supplements (see Table 1). These differences might result in conflicting findings, such as those of Hornyak et al. [33], who examined the effects of magnesium oxide supplementation in insomniacs and demonstrated (n = 10, by means of polysomnography) a 10% increase in sleep efficiency in parallel with superior total sleep times (+46 min pre–post treatment). However, such findings may have been influenced by a placebo effect due to the absence of a control with the design. More recently, a double-blind, placebo-controlled study conducted by Rondanelli et al. [34] concluded that 8 weeks of food supplementation (5 mg of melatonin, 225 mg of magnesium and 11.25 mg of zinc) significantly enhanced subjective sleep quality in elderly patients with severe insomnia. Whether this was due to the melatonin alone, magnesium or zinc supplementation (which are valuable for the endogenous synthesis of melatonin, ref. [35]) or the combined effects of the supplements is unclear [36]. That said, in a normal population that reports no sleep problems, exogenous melatonin supplementation has been shown to have no benefits on subjective measures of sleep assessment, regardless of time-zone travel, when arguably melatonin should have the greatest chance of aiding sleep [8,9].

The presence of vitamin B6 did not impact the factors associated with sleep (Table 2), despite its association with serotonin and melatonin production [37]. Even in studies where much larger doses (100 mg and 250 mg) of vitamin B6 were administered, there was no impact on melatonin levels or sleep parameters in healthy adults [16]. Observations that vitamin B6 [16] or multivitamins containing B6 [38,39] may influence sleep quality, reflected in subjective measurements of fatigue, were also identified in these participants (Table 3). Vitamin B6 appears to negatively impact sleep when delivered in higher (>100 mg) doses; however, the small level administered in ZMA is unlikely to have any effect on sleep or performance in participants with no medical sleep conditions. It is unclear if there may be any larger effect on those with disturbed sleep patterns.

The reported benefits of ZMA use to aid sleep in patients with severe insomnia who find it extremely difficult to initiate or maintain quality sleep [34] are incompatible with the current study. The cohort participating in the current study had baseline sleep data that showed no signs of sleep deprivation, where the parameters of sleep quality were similar to those of Olympic athletes (Table 2; [40]). The ineffectiveness of an acute dose of ZMA on young healthy male participants is apparent. However, the potential benefit of the acute use of ZMA on athletes on the one or two nights before competition when they anecdotally express poor sleep either due to nerves, stress of travel or sleeping in a foreign bed is unknown [9,41] and warrants further experimentation. 

The second main finding of the study was that ZMA supplementation had no beneficial effect on RSP times (*p* > 0.05), although CMJ height was lower in PLAC than in NoPill and ZMA conditions (*d* = 0.35, *p* < 0.05; Table 2). The reason for the negative placebo effect (of taking 1.55 g of glucose dextrose powder) is unclear. (i) Certainly, in the current study, the participants were fully familiarized with the performance measures, (ii) the order of conditions was counterbalanced to statistically distribute any residual learning effects and (iii) the participants did not know which tablet they were taking (as evidenced by their response to this question). In this study, just having a fully balanced placebo design as suggested by others without a NoPill control group would have changed the interpretation of the study and given the false positive that ZMA improves jump height [42].

The supplementation of ZMA has previously been reported to have no effect on muscular endurance or anaerobic power in 42 resistance-trained males [13], or a cohort of 12 experienced weightlifters [43] in submaximal weightlifting at 40, 60 and 80% 1RM for bench press and back squat [19]. However, in semi-professional football players, Brilla and Conte [44] demonstrated (via a Biodex isokinetic dynamometer) a 2.5 times greater increase in strength and functional power concomitant with substantial increases in testosterone levels following a 7-week supplementation period. Comparing the current study’s findings and those of others is difficult, as this study recruited participants of a lower training status than others, with different performance outcomes measured, such as the 10 × 20 m RSP protocol, which is most reflective of performance in a teamsport, based on several recent time–motion analysis studies [20,45]. It is worth noting that the current study used an acute rather than chronic supplementation protocol, as the multiple and complex effects of training and diet over days may have added a complication to any ergogenic effect [46]. Lastly, regardless of the condition (ZMA, PLAC or NoPill), there was no difference in the Stroop task either in the number of words completed or the errors associated with the task of reading. The basis of the Stroop effect relies on humans having trouble naming a physical color when it is used to spell the name of a different color, such that there is a delay in the reaction time between congruent and incongruent stimuli. Although the Stroop task has been shown to be affected by sleep loss, it could be that natural sleep loss (~2 h, Table 2) associated with an 8-h sleep opportunity was not enough of a stress that ZMA could have improved. 

Taken together, the current body of literature would suggest that the athletic benefit to performance of consuming ZMA is weak. Furthermore, although there is convincing evidence that zinc supplementation may reverse the negative effects of nutritional deficiencies [47] and aid pathological conditions in the elderly [48], this cannot be transferred directly to non-deficient and healthy exercising populations used in the current study, and probably the target population of the supplement industry.

## 5. Conclusions

To summarize, the findings in this investigation highlight the lack of ergogenic effects the dietary supplement ZMA offers young healthy male participants who have no diagnosed sleep disorders. ZMA had no effect on objective and subjective sleep in addition to fostering any support for next-day morning Stroop test (attention and notably its inhibitory processes) or physical performance measurements of repeated sprint and counter movement jump ability. Although recruitment met the identified power (n = 19), it may still be difficult to generalize to the population, therefore, further research with greater numbers is warranted. Furthermore, these novel findings highlight the importance of using both a PLAC and a NoPill group in ZMA research to elucidate the effects of the intervention under investigation.

## 6. Limitations

As there were no similar studies to take an effect size from for sample size estimation, we adopted a pragmatic approach. We chose a small effect size of 0.6 for our a priori sample size estimation, which predicted that a sample of 19 male participants was required. This may be an underestimation for this type of work. However, we followed the checklist of considerations in chronobiological studies on humans and sporting performance [28] to reduce bias (of measurement) from (i) participant considerations, (ii) methodological and equipment considerations, and (iii) environment considerations. In the design of the current study, we applied this advice to best reduce the signal-to-noise ratio to allow any effects of the supplementation, should it exist, to be found. The magnitude of an effect size from a similar study but with a different population, using less rigorous methods, may not be appropriate in sample size estimation for the current investigation. We used actimetry to assess sleep. This method lacks sensitivity to detect changes specifically in sleep latency, due to the device being unable to distinguish between movement during sleep and general non-movement. Although, polysomnography offers a greater level of accuracy in detecting meaningful change, this method requires added time, expertise, time demand on the participant and cost. As we chose a relatively small sample of young males, this limits the generalizability of the findings. Lastly, in the current investigation, females were omitted, and further research should investigate the potential sex differences in supplementation effects with or without sleep loss. It is suggested that not only sleep but also the full 24 h after waking up on night two are investigated so that the rhythms of mood, core body temperature, tiredness and alertness (to investigate phase-advanced profiles in women vs. men) as well as cognitive and physical performance are measured.

## Figures and Tables

**Figure 1 nutrients-16-02466-f001:**
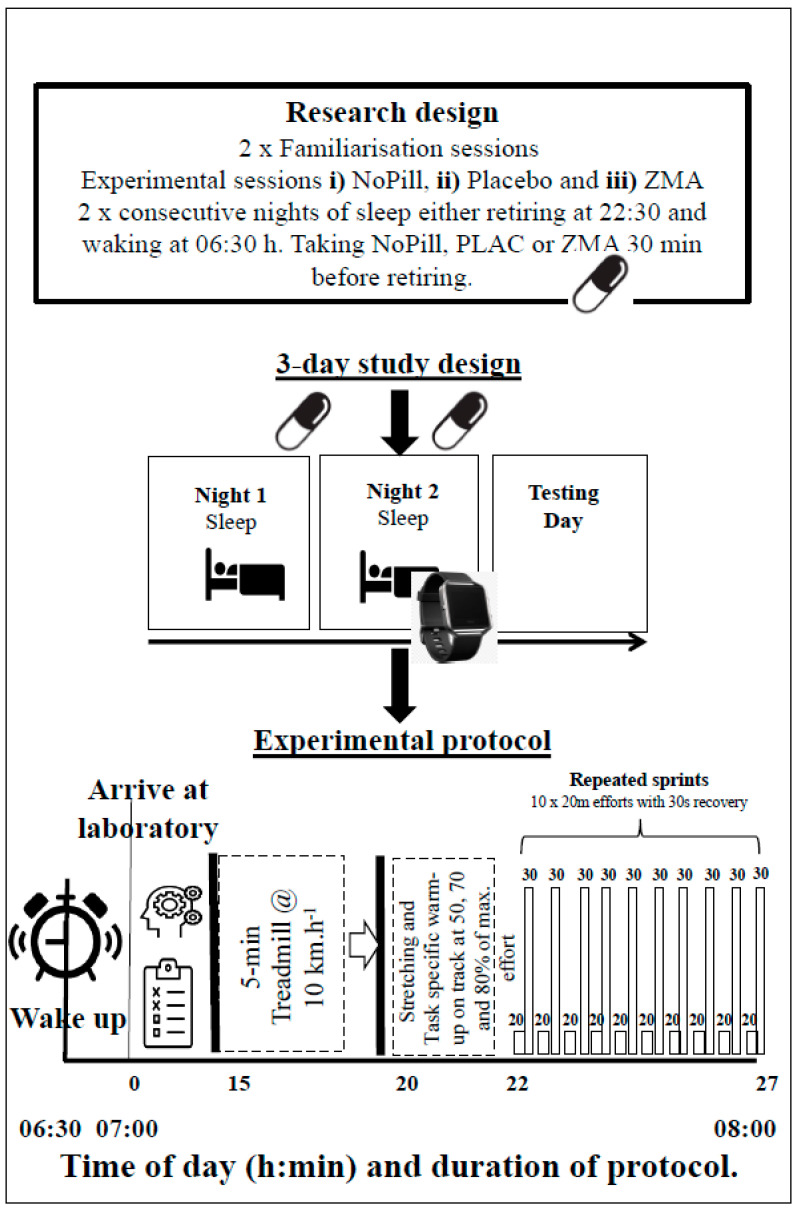
Schematic representation of the protocol undertaken by participants in the investigation. The watch represents Night 2 sleep recorded by actigraphy. The Stroop test and questionnaires are represented by head and clipboard, respectively.

**Figure 2 nutrients-16-02466-f002:**
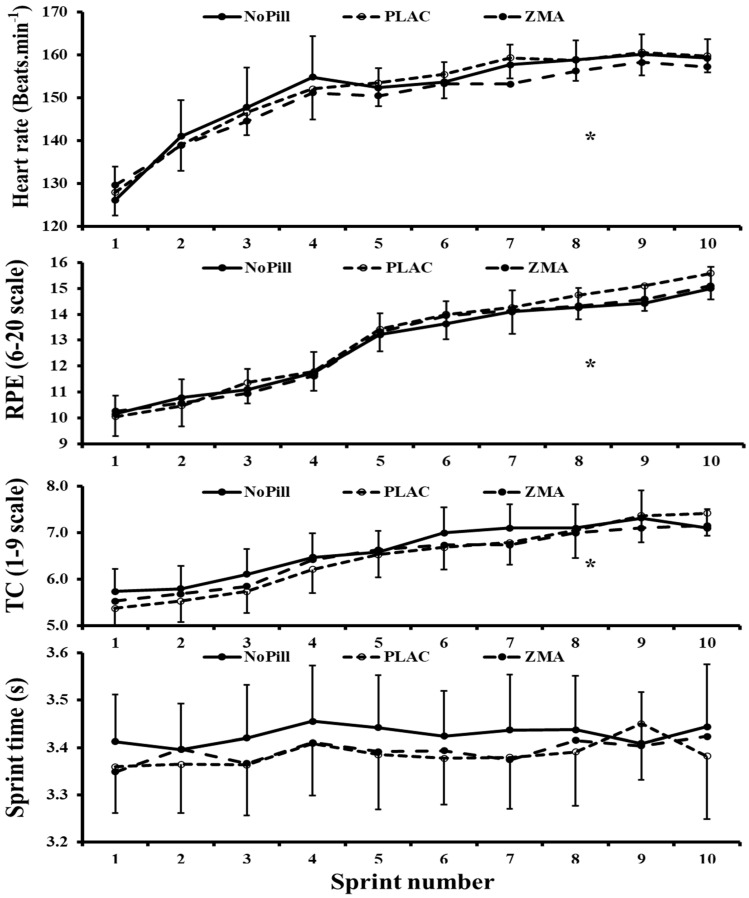
Mean (±95 CI) values for heart rate, rating of perceived exertion, thermal comfort and sprint time values every sprint (10 times, RSP) for NoPill, PLAC and ZMA conditions. * denotes the main effect for sprint number.

**Table 1 nutrients-16-02466-t001:** Physical characteristics, baseline actimetry and food intake of the participants (mean ± SD values). * denotes recommended daily values (RDA) from Nutrition Science Team [24]. **Bold** values denote significant participants’ values compared to RDA from a paired *t*-test.

	Participant Variables	
**Physical characteristics**		
Age (year)	24.6 ± 5	
Height (cm)	175.9 ± 7.5	
Mass (kg)	77.9 ± 9.7	
**Baseline actimetry**		
Fragmentation (Au)	29.0 ± 13.0	
Sleep efficiency (%)	82.0 ± 7.0	
Actual sleep time (h:mm)	6:44 ± 00:45	
Habitual retiring time (h:mm)	23:10 ± 00:31	
Habitual wake time (h:mm)	7:13 ± 00:38	
Sleep onset latency (h:mm)	0:12 ± 00:12	
Time in bed (h:mm)	22:30 ± 00:32	
Wake time (h:min)	07:10 ± 00:32	
**Baseline food intake**	**Participants intake**	**RDA ***
Daily calories (kcal)	2688 ± 632	2500 (*p* = 0.210)
Fats (g)	**153 ± 85**	**97 (*p* = 0.010)**
Protein (g)	**140 ± 45**	**55.5 (*p* < 0.001)**
Carbohydrates (g)	**188 ± 100**	**330 (*p* < 0.001)**
Zinc (mg)	11.9 ± 3.4	11 (*p* = 0.257)
Magnesium (mg)	395 ± 103	400 (*p* = 0.832)
Vitamin B6 (mg)	**2.7 ± 0.9**	**1.4 (*p* < 0.001)**

**Table 2 nutrients-16-02466-t002:** A comparison of the mean (±SD) values for actimetry, subjective sleep measures, repeated sprint (times, HR, thermal comfort, RPE) and CMJ performance, the Stroop test and blood variables measured in all conditions (NoPill, PLAC and ZMA). The main effect for condition, sprint/pre–post and interactions as well as Cohen’s *d* effect size and observed power are given. Statistical significance (*p* < 0.05) is indicated in **bold**, and a trend (where 0.10 > *p* > 0.05) is indicated in *italics*. QTN denotes questionnaire.

Variable	NoPill	PLAC	ZMA	Main Effect for		Main Effect Sprint/Pre-Post, *d* and Observed Power		Interaction
				Condition, *d* and Observed Power				
** *Actigraphy* **								
Sleep onset latency (h:mm)	0:20 ± 0:20	0:21 ± 0:27	0:15 ± 0:12	*p* = 0.532	0.35, 13.5			
Sleep efficiency (%)	80.0 ± 7.0	81.0 ± 8.0	79.0 ± 7.0	*p* = 0.408	0.30, 18.7			
Actual sleep time (h:mm)	6:09 ± 0:31	6:08 ± 0:31	6:00 ± 0:32	*p* = 0.558	0.24, 14.1			
Fragmentation index (%)	30.9 ± 11.5	28.1 ± 10.1	29.2 ± 13.8	*p* = 0.702	0.22, 97.0			
** *Subjective Sleep QTN* **								
** *Waterhouse QTN* **								
Ease to sleep	−0.7 ± 1.8	−0.7 ± 2.3	−1.1 ± 2.6	*p* = 0.812	0.17, 80.0			
Time to sleep	−0.6 ± 1.9	−0.2 ± 2.5	−0.5 ± 2.1	*p* = 0.808	0.14, 81.0			
How well did you sleep	−0.9 ± 1.9	−0.4 ± 2.2	−0.9 ± 2.0	*p* = 0.679	0.23, 10.8			
What was your waking time	−1.3 ± 2.3	−0.8 ± 1.9	−0.8 ± 1.6	*p* = 0.592	0.27, 100.0			
Alertness 30-min after waking	−0.3 ± 2.3	0.1 ± 1.5	0.2 ± 1.9	*p* = 0.672	0.06, 94.0			
** *Stanford Sleep QTN* **								
Degree of sleepiness	3.2 ± 0.7	2.9 ± 0.7	2.9 ± 1.1	*p* = 0.613	0.05, 11.6			
** *Cognitive Function* **								
Stroop (no. of mistakes)	4.2 ± 3.1	6.5 ± 1.2	6.5 ± 1.2	*p* = 0.609	0.28, 12.5			
Stroop (no. completed)	149.8 ± 24.1	150.1 ± 25.1	153.2 ± 30.5	*p* = 0.795	0.13, 83.0			
** *Jump (CMJ)* **								
Height (cm)	32.5 ± 6.2	30.7 ± 5.6 *	32.3 ± 5.7	***p <* 0.001**	**0.354, 97.5**			
** *Repeated Sprint Ability (RS)* **								
Split time (s)	3.43 ± 0.23	3.39 ± 0.25	3.39 ± 0.24	*p = 0.052*	*0.147, 57.0*	*p* = 0.217	0.176, 0.439	*p* = 0.162
RPE (6–20)	12.8 ± 2.3	13.1 ± 2.6	12.9 ± 2.6	*p* = 0.640	0.017, 11.7	***p* < 0.001**	**2.797, 100.0**	*p* = 0.354
TC (1–9)	6.6 ± 1.2	4.0 ± 2.1	3.8 ± 2.2	*p* = 0.454	0.122, 17.7	***p* < 0.001**	**1.707, 100.0**	*p* = 0.218
HR (beats.min^−1^)	151 ± 18	151 ± 17	149 ± 14	*p* = 0.543	0.108, 14.6	***p* < 0.001**	**2.560, 100.0**	*p* = 0.752
** *Blood* **								
Lactate Pre (mmol L^−1^)	0.9 ± 0.2	1.1 ± 0.4	1.0 ± 0.3	*p = 0.075*	*0.199, 51.3*	***p* < 0.005**	**3.998, 100.0**	*p = 0.081*
Lactate Post (mmol L^−1^)	5.7 ± 2.2	5.7 ± 1.9	6.2 ± 2.2					

(CMJ, counter movement jump; RPE, rating of perceived exertion; TC, thermal comfort; HR, heart rate; * *p* < 0.001).

**Table 3 nutrients-16-02466-t003:** Mean (± SD) values for perceived onset of mood scores (POMS) of conditions (NoPill, PLAC and ZMA) at two times of the day, with p-values given. Statistical significance (*p* < 0.05) is indicated in **bold**, and a trend (where 0.10 > *p* > 0.05) is indicated in *italics*.

Variable	NoPill		PLAC		ZMA		CON	Time-of-Day	Interaction
	Pre-Sleep (30-min)	After Wake (30-min)	Pre-Sleep (30-min)	After Wake (30-min)	Pre-Sleep (30-min)	After Wake (30-min)			
** *POMS* **									
*Anger*	0.9 ± 1.7	0.7 ± 1.7	1.2 ± 2.8	0.6 ± 1.5	1.3 ± 2.1	1.2 ± 2.3	*p* = 0.682	*p* = 0.111	*p* = 0.786
*Depression*	1.1 ± 2.1	0.7 ± 1.7	1.1 ± 2.1	0.5 ± 1.1	0.8 ± 1.5	0.8 ± 1.5	*p* = 0.900	*p = 0.052*	*p* = 0.484
*Vigour*	4.2 ± 3.3	5.0 ± 3.1	5.4 ± 3.5	5.8 ± 3.3	4.9 ± 3.1	5.3 ± 2.6	*p* = 0.234	*p* = 0.371	*p* = 0.878
*Calm*	8.6 ± 3.9	7.5 ± 3.6	8.4 ± 2.9	6.3 ± 3.1	7.5 ± 3.9	6.7 ± 2.8	*p* = 0.465	***p* = 0.005**	*p* = 0.295
*Tension*	0.3 ± 0.9	0.3 ± 0.8	0.5 ± 1.2	0.4 ± 0.8	0.7 ± 1.4	0.1 ± 0.2	*p* = 0.610	*p = 0065*	***p* = 0.036**
*Confusion*	1.3 ± 1.9	0.9± 2.0	1.5 ± 2.6	1.0 ± 1.6	1.2 ± 1.9	0.7 ± 1.6	*p* = 0.713	***p* = 0.002**	*p* = 0.934
*Fatigue*	7.3 ± 2.2	5.9 ± 1.8	6.8 ± 2.2	4.4 ± 2.2	5.9 ± 2.7	5.1 ± 2.5	***p* = 0.022**	*p* = 0.001	***p* = 0.016**
*Happy*	8.1 ± 3.7	5.6 ± 2.8	7.4 ± 3.4	6.2 ± 2.9	6.6 ± 3.4	5.3 ± 3.0	*p* = 0.223	***p* = 0.001**	*p* = 0.347

(POMS = Profile of Mood State).

## Data Availability

The data are not publicly available due to ethical restrictions.

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
