# Peer review of "Effects of an Acute Dose of Zinc Monomethionine Asparate and Magnesium Asparate (ZMA) on Subsequent Sleep and Next-Day Morning Performance (Countermovement Jumps, Repeated Sprints and Stroop Test)"

_nutrients, 2024, doi:10.3390/nu16152466_

Round 1

Reviewer 1 Report

Comments and Suggestions for Authors

Abstract:

11-12: Improve summary of the Background section.

52-61: The authors associate ZMA supplementation with improved sleep quality. I recommend delving deeper into the physiological mechanisms that justify such a statement. I suggest introducing the physiological variable lactate, since the research team analyzed it.

52-61: The authors assume polysomnography as the gold standard to assess sleep quality, but in their work they only assessed activity with actiwatch and sleep with questionnaires. Justify

64: The authors speak of a “large sample”, the number of this study was 19 participants. I recommend using "significant sample".

Table 1: I recommend adding it to the material and methods section, participants subsection.

74-77: Justify how with such a small sample size significant data can be obtained using the test they propose.

91-92: The study was approved by the committee of which institution/University?

95: The type of study that has been carried out is not specified, whether it was observational, descriptive, analytical...

105-106: I recommend explaining the characteristics of the questionnaires, what do they consist of? Are they self-administered or was it administered by a team member? What do they measure?

159-181: How do the authors know the previous values â€‹â€‹related to the sleep quality of the participants? Were the questionnaires administered pre and post intervention?

Table 2: I suggest it be added to the results section.

Discussion

- I suggest trying to evaluate the findings with similar studies or with similar measurement instruments.

I recommend adding limitations section after discussion.

Conclusions:

- I recommend that the conclusions respond to the objective set and specify the effects of both the placebo PLAC group or without the NoPill  in relation to sleep and performance.

Therefore, I recommend that if it is an observational study, the STROBE recommendations or guidelines for this type of research should be followed.

Author Response

Q 1. 11-12: Improve summary of the Background section.

A1. We thank the reviewer for their question, we agree and have added to the introduction regarding sleep, the mechanism of action proposed for ZMA to improve sleep and what the Stroop task is, and its purpose. See lines 38-75. We have changed a large portion of the introduction to strengthen the background summary– see below.

Lines 61-82: Investigations into the effectiveness of supplementation of Zn and Mg have focused on populations whose daily habitual intake are below recommended levels, are either very young or aged and are clinical based such as insomniacs [18,19]. Methodological issues have limited interpretation of this area such as a lack of i) prior measurement of diet for macro and micronutrient quantities, ii) baseline assessment of habitual sleep and resultant sleep after ingestion of the supplement; as well as iii) the absence of a “Pill” condition, as well as a Placebo (PLAC) and experimental (ZMA) condition. Recent work that addressed the concerns about research design has investigated if recreational athletes (n=16) suffering from partial sleep deprivation (but are otherwise healthy with no sleep disorders with a balanced diet), may experience improvements in sleep variables with acute ZMA supplementation [20]. No benefits of supplementation on sleep (objective or subjectively measured) nor morning Stroop performance were found. Further, research investigating acute ZMA supplementation (1-2 nights) is therefore warranted in populations of “normal sleepers” with balanced diets who through an 8-h opportunity of sleep may have sleep loss, to further understand effects on sleep and morning cognitive performance and other physical performance measures. One such measure is repeated sprint ability (RSA) the ability to repeatedly perform maximal sprints (≤6 s) with limited recovery between bouts (≤60 s) is an important performance test as it carries relevance for team-based sports; but is also sensitive to environmental change such as time-of-day [21].  

Q2. 52-61: The authors associate ZMA supplementation with improved sleep quality. I recommend delving deeper into the physiological mechanisms that justify such a statement. I suggest introducing the physiological variable lactate, since the research team analyzed it.

A2. We thank the reviewer for their question and have added to the Introduction regarding the proposed for ZMA to improve sleep – see below.

Lines 34-41. Nutritional supplement companies claim that the exogenous supplementation of Zinc monomethionine Asparate and Magnesium Asparate (ZMA) will promote a deep and restful sleep (due to magnesium's ability to normalize and extend stage 3 and stage 4 slow-wave sleep), aid in circadian regulation and increase testosterone levels [3], thereby aiding recovery and increasing muscular strength. It has recently been proposed that the mechanism in which Zn and Mg can promote sleep is through their roles in synthesis and function of sleep–wake neurotransmitters such as gamma-aminobutyric acid (GABA)—a receptor that supports sleep when activated [4].

Q3. 52-61: The authors assume polysomnography as the gold standard to assess sleep quality, but in their work they only assessed activity with actiwatch and sleep with questionnaires. Justify

A3. We thank the reviewer for their question we have taken this out of the introduction and considered this in the new limitation section of the manuscript. See below.

Lines 328-331: We used actimetry to assess sleep, this method lacks sensitivity to detect changes specifically in sleep latency, due to the device being unable to distinguish the difference between movement during sleep and general non-movement. Although, polysomnography offers a greater level of accuracy in detecting meaningful change, however this method requires added time, expertise, time demand on the participant and cost.

Q4. 64: The authors speak of a “large sample”, the number of this study was 19 participants. I recommend using "significant sample".

A4. We thank the reviewer for their comment and have removed “large” from the text.

Lines 76-82 Therefore, the aims of this study were to investigate the effects of an acute dose of ZMA [compared to a placebo (PLAC) or no pill condition (NoPill)] taken before retiring for two nights (with the opportunity to sleep for 8-h), on subsequent sleep and next day morning physical (RPA and Countermeasure jumps) and Stroop (attention and notably its inhibitory processes) performance in a sample of familiarized and highly motivated participants, with no known sleep disorders. We hypothesized that the ZMA would have no beneficial effects on sleep variables or next day morning physical or Stroop performance in our chosen population of healthy male recreational athletes following 2 nights of supplementation.

Q5. Table 1: I recommend adding it to the material and methods section, participants subsection.

A5. We thank the reviewer for their comment, Table 1 is associated with text in the Materials and Methods section we had just put it after the aims for convenience – we have moved this now (page 3) to avoid confusion. See text below where Table 1 is mentioned.

Lines 92: Baseline data of the participants is presented in Table 1

Q6. 74-77: Justify how with such a small sample size significant data can be obtained using the test they propose.

A6. We thank the reviewer for their question, this is an interesting point. I am assuming the question is relating to the use of the effect size for sample size estimation. There are no previous studies that we could use to calculate an effect size that we could subsequently use for sample size estimation. Consequently, we had to consider the effect size that would be meaningful and realistic; the effect size we’d be interested in accepting / rejecting. Considering the effect size scale proposed by Batterham and Hopkins [32] (please see below re justification about the scale), we deemed a value of 0.6, which was at the higher end of a small effect was appropriate. This decision was because we expected a small effect from the intervention and our sample being recreationally active participants.

Lines 184-185…. The magnitude of the ES was classified as trivial (≤0.2), small (>0.2–0.6), moderate (>0.6–1.2), large (>1.2–2.0) and very large (>2.0) based on guidelines from Batterham and Hopkins [32].

Please note in the current investigation we have followed the checklist of considerations in chronobiological studies on humans and sporting performance (Edwards et al. 2024; https://doi.org/10.1080/09291016.2024.2316401) to reduce bias (of measurement) from i) participant considerations, ii) methodological and equipment considerations, iii) environment considerations. In the design of the current study, we applied this advice to best reduce the signal to noise ratio, to allow any effects of the supplementation should it exist, to be found. The magnitude of an effect size from a similar study but with a different population, using less rigorous methods may not be appropriate in sample size estimation for the current investigation.

We have put a section in the limitations regarding effect size – see below.

Lines 322-329: As there were no similar studies to take an effect size from for sample size estimation, we adopted a pragmatic approach. We chose a small effect size of 0.6 for are a’priori sample size estimation which predicted a sample of 19 male participants were required. This may be an underestimation for this type of work. However, we followed the checklist of considerations in chronobiological studies on humans and sporting performance [28] to reduce bias (of measurement) from i) participant considerations, ii) methodological and equipment considerations, and iii) environment considerations. In the design of the current study, we applied this advice to best reduce the signal to noise ratio, to allow any effects of the supplementation should it exist, to be found. The magnitude of an effect size from a similar study but with a different population, using less rigorous methods may not be appropriate in sample size estimation for the current investigation.

Q7. 91-92: The study was approved by the committee of which institution/University?

A7. We thank the reviewer for their comment we have inserted the University name in the text now – see below.

Lines 99-100: Experimental procedures were approved by Liverpool John Moores University Human Ethics Committee (P14SEC018) and conducted in accordance with the ethical standards of the journal and complied with the Declaration of Helsinki.

Q8. 95: The type of study that has been carried out is not specified, whether it was observational, descriptive, analytical...

A8. We thank the reviewer for their comment this is an experimental/interventional study as we conducted the experiment with interventions.

Q9. 105-106: I recommend explaining the characteristics of the questionnaires, what do they consist of? Are they self-administered or was it administered by a team member? What do they measure?

A9. We thank the reviewer for their comment, we agree and have added to the information regarding the questionnaire – see below.

Lines 164-169: Following two consecutive nights of sleep and normal activity the participants rose at 6:30, arrived at the laboratory at 07:00 h and after sitting for 30 mins they completed the rating of Profile of Mood State questionnaire (POMS [25]), sleep questionnaires and sleepiness [26,27]. The POMS questionnaire measures mood states with the participants answering 32 questions with 8 moods for how they “feel right now”. For the sleep questions participants ticked on a 10 cm scale their answers “compared to normal” with -5 “being less or earlier”, 0 “normal” and +5 “more or later”.

Q10. 159-181: How do the authors know the previous values â€‹â€‹related to the sleep quality of the participants? Were the questionnaires administered pre and post intervention?

A10. We thank the reviewer for their comment, the sleep questions from the Waterhouse et al. work are normalised too normal. We have added this to the text to remove this confusion – see below.

Lines 164-169: Following two consecutive nights of sleep and normal activity the participants rose at 6:30, arrived at the laboratory at 07:00 h and after sitting for 30 mins they completed the rating of Profile of Mood State questionnaire (POMS [25]), sleep questionnaires and sleepiness [26,27]. The POMS questionnaire measures mood states with the participants answering 32 questions with 8 moods for how they “feel right now”. For the sleep questions participants ticked on a 10 cm scale their answers “compared to normal” with -5 “being less or earlier”, 0 “normal” and +5 “more or later”.

Q11. Table 2: I suggest it be added to the results section.

A11. We thank the reviewer for their comment, we agree and have put this in the results section.

Q12. Discussion - I suggest trying to evaluate the findings with similar studies or with similar measurement instruments.

A12. We thank the reviewer for their comment, we agree and where possible with the challenge of sparce literature in this area.

Q13. I recommend adding limitations section after discussion.

A13. We thank the reviewer for their comment, we agree and have included a limitations section – see below.

Lines 320-335: 6. Limitations

As there were no similar studies to take an effect size from for sample size estimation, we adopted a pragmatic approach. We chose a small effect size of 0.6 for are a’priori sample size estimation which predicted a sample of 19 male participants were required. This may be an underestimation for this type of work. However, we followed the checklist of considerations in chronobiological studies on humans and sporting performance [28] to reduce bias (of measurement) from i) participant considerations, ii) methodological and equipment considerations, and iii) environment considerations. In the design of the current study, we applied this advice to best reduce the signal to noise ratio, to allow any effects of the supplementation should it exist, to be found. The magnitude of an effect size from a similar study but with a different population, using less rigorous methods may not be appropriate in sample size estimation for the current investigation. We used actimetry to assess sleep, this method lacks sensitivity to detect changes specifically in sleep latency, due to the device being unable to distinguish the difference between movement during sleep and general non-movement. Although, polysomnography offers a greater level of accuracy in detecting meaningful change, however this method requires added time, expertise, time demand on the participant and cost. As we chose a young population of males with a relatively small sample size this limits the generalizability of the findings. Lastly, in the current investigation females were omitted, further research should investigate the potential sex difference in supplementation effects with or without sleep loss. It is suggested that as well as sleep the full 24-h after waking on night two are investigated so the rhythms of mood, core body temperature tiredness and alertness (to investigate phase advanced profiles in female's vs men) as well as cognitive and physical performance are measured.Q13. Conclusions: - I recommend that the conclusions respond to the objective set and specify the effects of both the placebo PLAC group or without the NoPill in relation to sleep and performance.

A13. We thank the reviewer for their comment, we agree and have amended the text accordingly – see below.

Lines 313-319: To summarise, the findings in this investigation highlight the lack of ergogenic effects the dietary supplement ZMA offers young healthy male participants, who have no diagnosed sleep disorders. The ZMA had no effect on objective and subjective sleep, in addition to foster any support for next day morning Stroop (attention and notably its inhibitory processes), or physical performance measurements of repeated sprint and counter movement jump ability. Although recruitment met the identified power (n=19), it may still be difficult to generalise to the population, therefore further research with greater numbers is warranted. Furthermore, these novel findings highlight the importance of using both a PLAC and a NoPill group in ZMA research to elucidate the effects of the intervention under investigation.

Q14. Therefore, I recommend that if it is an observational study, the STROBE recommendations or guidelines for this type of research should be followed.

A14. We thank the reviewer for their comment this is an experimental/interventional study as we conducted the experiment with interventions.

Reviewer 2 Report

Comments and Suggestions for Authors

Excellent research, only some questions

Refeer where you found the effect to calculate yoour sample size

You should perform post hoc pairwise analysis wiith Bonferroni correction to find between specific conditions ant tima are differences

Author Response

Q1. Excellent research, only some questions

Refeer where you found the effect to calculate yoour sample size

A1. We thank the reviewer for their question, this is an interesting point. I am assuming the question is relating to the use of the effect size for sample size estimation. There are no previous studies that we could use to calculate an effect size that we could subsequently use for sample size estimation. Consequently, we had to consider the effect size that would be meaningful and realistic; the effect size we’d be interested in accepting / rejecting. Considering the effect size scale proposed by Batterham and Hopkins [32] (please see below re justification about the scale), we deemed a value of 0.6, which was at the higher end of a small effect was appropriate. This decision was because we expected a small effect from the intervention and our sample being recreationally active participants.

Lines 184-185…. The magnitude of the ES was classified as trivial (≤0.2), small (>0.2–0.6), moderate (>0.6–1.2), large (>1.2–2.0) and very large (>2.0) based on guidelines from Batterham and Hopkins [32].

Please note in the current investigation we have followed the checklist of considerations in chronobiological studies on humans and sporting performance (Edwards et al. 2024; https://doi.org/10.1080/09291016.2024.2316401) to reduce bias (of measurement) from i) participant considerations, ii) methodological and equipment considerations, iii) environment considerations. In the design of the current study, we applied this advice to best reduce the signal to noise ratio, to allow any effects of the supplementation should it exist, to be found. The magnitude of an effect size from a similar study but with a different population, using less rigorous methods may not be appropriate in sample size estimation for the current investigation.

We have put a section in the limitations regarding effect size – see below.

Lines 322-329: As there were no similar studies to take an effect size from for sample size estimation, we adopted a pragmatic approach. We chose a small effect size of 0.6 for are a’priori sample size estimation which predicted a sample of 19 male participants were required. This may be an underestimation for this type of work. However, we followed the checklist of considerations in chronobiological studies on humans and sporting performance [28] to reduce bias (of measurement) from i) participant considerations, ii) methodological and equipment considerations, and iii) environment considerations. In the design of the current study, we applied this advice to best reduce the signal to noise ratio, to allow any effects of the supplementation should it exist, to be found. The magnitude of an effect size from a similar study but with a different population, using less rigorous methods may not be appropriate in sample size estimation for the current investigation.

Q2. You should perform post hoc pairwise analysis wiith Bonferroni correction to find between specific conditions ant tima are differences

A2. We thank the reviewer for their question, we agree and we used Bonferroni pairwise comparisons see text below.

Lines 182-183. Graphical comparisons between means and Bonferroni pairwise comparisons were made where main effects were present.

Reviewer 3 Report

Comments and Suggestions for Authors

Thank you for submitting your work for review.

In the initial analysis, I noticed a significant issue: the work relies on outdated literature, with 76% of the referenced publications being older than 10 years.

The bibliography is written in various styles, which significantly hinders the readability of the work. It needs to be corrected according to the journal’s guidelines.

Specific points:

  • L7-9 – There is a different font here; please correct it.
  • L67 – "Table 1." – There should be a reference first, followed by the table.
  • Add a research hypothesis at the end of the introduction.
  • L76 – Why was an effect size of 0.6 chosen? Please justify.
  • L193 – Why did the authors choose this specific effect size extension rather than the one proposed by Sawilowsky, for example?
  • Table 2 – Again, there should be a reference first, followed by the table.
  • In each of the tables, authors should correct: the authors present many values in columns, such as "Main effect for condition, d and observation power," which significantly hinders readability. Split these into separate columns.
  • Additionally, standardize the format of numerical values. Sometimes they are given to two decimal places, other times to three. Please make them consistent.
  • Under each table, expand all the abbreviations used in the tables.
  • The conclusions need to be completely revised. They largely repeat the results. Additionally, emphasize that the study needs to be improved with a larger sample size. N=19 is difficult to generalize to the population.

Author Response

Reviewer 3

Q1. Thank you for submitting your work for review.

In the initial analysis, I noticed a significant issue: the work relies on outdated literature, with 76% of the referenced publications being older than 10 years.

A1. Thank you for your comments. You raise a good point, indeed the majority of the research in this area is >10 years old. here is a lot of older research, but we feel it is important to recognise the excellent work before us and provide this background. These older studies are fundamental to this area and provide the footprint to follow. There may be less now, but we still feel this is an important area of research to explore further and gaps in the literature as we have identified. 

Q2. The bibliography is written in various styles, which significantly hinders the readability of the work. It needs to be corrected according to the journal’s guidelines.

A1. We thank the reviewer for their comment and have corrected the reference section to the journal Nutrients guidelines.

Q3. Specific points: L7-9 – There is a different font here; please correct it.

A3. Amended to Palatino Lino type font.

Q4. L67 – "Table 1." – There should be a reference first, followed by the table.

A4. We thank the reviewer for their comment and have added the reference number [17] instead of the 2006 year. See below

Line 102-103. Table 1. Physical characteristics, baseline actimetry and food intake of the participants (mean ±SD values). * Denotes recommended daily values (RDA) from Nutrition Science Team [24]. Bold values denote significant of participants values compared to RDA from a paired t-test.

Q5. Add a research hypothesis at the end of the introduction.

A5. Thanks for highlighting this, we have added the following at the end of the introduction: Lines 61-62. We hypothesized that the ZMA would have no beneficial effects on sleep variables or next day morning physical or stroop performance in our chosen population of healthy male recreational athletes following 2 nights of supplementation.

Q6. L76 – Why was an effect size of 0.6 chosen? Please justify.

A6. We thank the reviewer for their question, this is an interesting point. I am assuming the question is relating to the use of the effect size for sample size estimation. There are no previous studies that we could use to calculate an effect size that we could subsequently use for sample size estimation. Consequently, we had to consider the effect size that would be meaningful and realistic; the effect size we’d be interested in accepting / rejecting. Considering the effect size scale proposed by Batterham and Hopkins [32] (please see below re justification about the scale), we deemed a value of 0.6, which was at the higher end of a small effect was appropriate. This decision was because we expected a small effect from the intervention and our sample being recreationally active participants.

Lines 184-185…. The magnitude of the ES was classified as trivial (≤0.2), small (>0.2–0.6), moderate (>0.6–1.2), large (>1.2–2.0) and very large (>2.0) based on guidelines from Batterham and Hopkins [32].

Please note in the current investigation we have followed the checklist of considerations in chronobiological studies on humans and sporting performance (Edwards et al. 2024; https://doi.org/10.1080/09291016.2024.2316401) to reduce bias (of measurement) from i) participant considerations, ii) methodological and equipment considerations, iii) environment considerations. In the design of the current study, we applied this advice to best reduce the signal to noise ratio, to allow any effects of the supplementation should it exist, to be found. The magnitude of an effect size from a similar study but with a different population, using less rigorous methods may not be appropriate in sample size estimation for the current investigation.

We have put a section in the limitations regarding effect size – see below.

Lines 322-329: As there were no similar studies to take an effect size from for sample size estimation, we adopted a pragmatic approach. We chose a small effect size of 0.6 for are a’priori sample size estimation which predicted a sample of 19 male participants were required. This may be an underestimation for this type of work. However, we followed the checklist of considerations in chronobiological studies on humans and sporting performance [28] to reduce bias (of measurement) from i) participant considerations, ii) methodological and equipment considerations, and iii) environment considerations. In the design of the current study, we applied this advice to best reduce the signal to noise ratio, to allow any effects of the supplementation should it exist, to be found. The magnitude of an effect size from a similar study but with a different population, using less rigorous methods may not be appropriate in sample size estimation for the current investigation.

Q7. L193 – Why did the authors choose this specific effect size extension rather than the one proposed by Sawilowsky, for example?

A7. We thank the reviewer for the question. The choice of Cohen’d measure of effect size rather than Glass’s delta or Hedges’g as the “groups/conditions” are the same participants, have similar standard deviations and are of the same size. The from Batterham and Hopkins reported scale was preferred over other effect size interpretation scales as it is commonly used in several related studies and allows a broader interpretation of the effect seen. We of course appreciate there are other similar scales, but the above reasons tipped the balance for that one.

Lines 184-185. The magnitude of the ES was classified as trivial (≤0.2), small (>0.2–0.6), moderate (>0.6–1.2), large (>1.2–2.0) and very large (>2.0) based on guidelines from Batterham and Hopkins [32].

Q8. Table 2 – Again, there should be a reference first, followed by the table.

A8. We thank the reviewer for their comment, we cannot see the need for a reference in the heading from lines 170-173?

Q9. In each of the tables, authors should correct: the authors present many values in columns, such as "Main effect for condition, d and observation power," which significantly hinders readability. Split these into separate columns.

A9. We thank the reviewer for their suggestion and have split the Effect size and power from the main effect for the general linear model.

Q10. Additionally, standardize the format of numerical values. Sometimes they are given to two decimal places, other times to three. Please make them consistent.

A10. We thank the reviewer for their comment. We have presented the variables to the limit of the accuracy of the measure so actigraphy can measure to the minute so h:mm may be 0:20, similarly split times are measured in seconds (s) to 2 decimal places as the measure is measured in m/s. We hope this answers your question.

Q11. Under each table, expand all the abbreviations used in the tables.

A11. We have added the following to the relevant tables:

Line 175: Table 2: (CMJ=counter movement jump; RPE=rating of perceived exertion; TC=thermal comfort; HR=heart rate; * p < 0.001.)

Line 215: Table 3: (POMS = profile of mood state)

Q12. The conclusions need to be completely revised. They largely repeat the results. Additionally, emphasize that the study needs to be improved with a larger sample size. N=19 is difficult to generalize to the population.

A12. Thanks for your comment. We have amended the conclusion to read as follows:

Lines 313-319: To summarise, the findings in this investigation highlight the lack of ergogenic effects the dietary supplement ZMA offers young healthy male participants, who have no diagnosed sleep disorders. The ZMA had no effect on objective and subjective sleep, in addition to foster any support for next day morning Stroop (attention and notably its inhibitory processes), or physical performance measurements of repeated sprint and counter movement jump ability. Although recruitment met the identified power (n=19), it may still be difficult to generalise to the population, therefore further research with greater numbers is warranted. Furthermore, these novel findings highlight the importance of using both a PLAC and a NoPill group in ZMA research to elucidate the effects of the intervention under investigation. 

Round 2

Reviewer 1 Report

Comments and Suggestions for Authors

thanks for your corrections. I consider that the manuscript meets the conditions for publication

Author Response

We thank the reviewer for their time and comments on this research - it is much appreciated.

Reviewer 3 Report

Comments and Suggestions for Authors

I accept in current forme.

Author Response

(The authors gave the same response as above.)
